# Oxytree Pruned Biomass Torrefaction: Mathematical Models of the Influence of Temperature and Residence Time on Fuel Properties Improvement

**DOI:** 10.3390/ma12142228

**Published:** 2019-07-10

**Authors:** Kacper Świechowski, Marek Liszewski, Przemysław Bąbelewski, Jacek A. Koziel, Andrzej Białowiec

**Affiliations:** 1Institute of Agricultural Engineering, Faculty of Life Sciences and Technology, Wrocław University of Environmental and Life Sciences, 37/41 Chełmońskiego Str., 51-630 Wrocław, Poland; 2Institute of Agroecology and Plant Production, Faculty of Life Sciences and Technology, Wrocław University of Environmental and Life Sciences, 24A pl. Grunwaldzki Str., 53-363 Wrocław, Poland; 3Department of Horticulture, Faculty of Life Sciences and Technology, Wrocław University of Environmental and Life Sciences, 24A pl. Grunwaldzki Str., 53-363 Wrocław, Poland; 4Department of Agricultural and Biosystems Engineering, Iowa State University, Ames, IA 50011, USA

**Keywords:** biorenewable energy, pruning biomass, torrefaction, torrefied biomass, fuel properties, Oxytree, model

## Abstract

Biowaste generated in the process of Oxytree cultivation and logging represents a potential source of energy. Torrefaction (a.k.a. low-temperature pyrolysis) is one of the methods proposed for the valorization of woody biomass. Still, energy is required for the torrefaction process during which the raw biomass becomes torrefied biomass with fuel properties similar to those of lignite coal. In this work, models describing the influence of torrefaction temperature and residence time on the resulting fuel properties (mass and energy yields, energy densification ratio, organic matter and ash content, combustible parts, lower and higher heating values, CHONS content, *H:C* and *O:C* ratios) were proposed according to the Akaike criterion. The degree of the models’ parameters matching the raw data expressed as the determination coefficient (R^2^) ranged from 0.52 to 0.92. Each model parameter was statistically significant (*p* < 0.05). Estimations of the value and quantity of the produced torrefied biomass from 1 Mg of biomass residues were made based on two models and a set of simple assumptions. The value of torrefied biomass (€123.4·Mg^−1^) was estimated based on the price of commercially available coal fuel and its lower heating value (*LHV*) for biomass moisture content of 50%, torrefaction for 20 min at 200 °C. This research could be useful to inform techno-economic analyses and decision-making process pertaining to the valorization of pruned biomass residues.

## 1. Introduction

The energy demand continues to increase, and researchers continue to develop alternative sources of energy. European Union directives aim to increase the share of renewable energy sources (RES) while lowering overall environmental impact. Renewable energy sources can have a positive impact on the environment and diversify energy supply. To date, ~10% of the total primary energy supply (TPES) is derived from biomass on a global scale [1]. The EU aims to increase the biomass share in the RES up to 50% [1]. By 2050, the share of RES in total energy consumption is expected to increase to 55% to 75% [2]. Thus, the demand for RES, including wood-based biomass is expected to grow. At present, it is not feasible to completely replace fossil fuels with RES in a sustainable manner. There are concerns about the negative impact of increased energy demand from biomass on biodiversity and food security [3]. However, introducing different biomass as feedstock could improve the biodiversity of energy crops. It is expected that the increase in the share of RES in the EU will lead to an increase in the demand for biomass from trees, which will lead to an increase in forested areas and short-rotation plantations [4].

Oxytree (*Paulownia Clon* in Vitro *112*) has been considered as a relatively new plant suitable for short rotation because of its quick-growing characteristics and the ability to produce a significant amount of biomass. Oxytree biomass yield increases significantly in a relatively short time. For example, the dry mass of the tree can increase tenfold from 0.21 to 2.05 kg d.m. from the first to the second year since planting [5]. Besides rapid growth, Oxytree is also more versatile than other energy crops. Oxytree’s wood can be used as a non-construction building material for paper, furniture, instruments, and others [6]. This versatility of end users is of great importance in case of an unexpected drop in the demand for bioenergy; it also allows greater flexibility in meeting the needs of the energy and industrial sectors.

The Oxytree biomass yield depends on many factors, such as stocking density and climate. With estimated stocking of 3300 trees per hectare, the yield in the 5-y period can amount to 80 Mg·ha^−1^ d.m., on average ~16 Mg·ha^−1^ d.m. per year [7]. Warm climates favored by paulownia can produce ~7.2–14.0 Mg·ha^−1^ d.m., with a planting density on a 3 m × 2 m grid, in conversion 1666 ha^−1^ per hectare and 6000 m^3^·ha^−1^ of irrigation in Andalusia [8].

Oxytree residues can be additionally utilized for energy purposes, similarly to the concept proposed by Dyjakon [9] for clippings from the apple orchard. The volume of plantation residues can be ~107 m^3^ per hectare assuming that clippings consist of ~70% of the total tree volume and that ~250 m^3^ of industrial wood can be obtained from 1 ha [10]. The application of torrefaction for the valorization of residual biomass fuel properties may increase the profitability and sustainability of energy production from the Oxytree biomass (Figure 1). To date, pruned biomass is typically left on the field, burned or composted on-site.

Torrefaction, a.k.a. ‘roasting’ or ‘mild pyrolysis,’ is a thermochemical process with a limited amount of oxygen at ~near atmospheric pressure. The biomass is torrefied at a temperature of 200–300 °C at most up to 1 h. The purpose of the process is to obtain a material (called torrefied biomass) that has improved fuel properties compared with the substrate used for its production. During the biomass torrefaction, gases such as H_2_, CO_2_, CO, CH_4_, C_x_H_y_, toluene, and benzene are produced in addition to steam, volatile organic compounds, and lipids. During the torrefaction, up to 30% of the mass losses occur while maintaining 90% of the energy content of the substrate. These mass losses result from volatilization of condensable and non-condensable gases products [11,12]. Biomass is pre-treated to produce a high-quality solid biofuel that can be used for combustion or gasification. Torrefaction is based on oxygen removal and decomposing of the reactive hemicellulose using temperature. The quality of obtained solid fuel depends on reaction conditions, such as temperature, inert gas type, reaction time, used feedstock, and others [13]. Although the torrefaction process has been known for a long time, it has only recently become popular again because of the commencement of co-combustion of coal with biomass in some power plants. The torrefied biomass has a higher energy value; it contains less moisture, and it is easier to grind compared with raw biomass. By subjecting the biomass to torrefaction, it is possible to obtain torrefied biomass with fuel properties similar to lignite coal [14]. The energy requirements related to grinding decreased up to ten times (from around 250 to 25 kWh∙Mg^−1^) for forest logging residues [15].

Pyrolyzed biomass is considered as an eco-friendly fuel that could reduce greenhouse gas emissions by sequestrating of atmospheric carbon into the soil. Torrefied biomass or biochar used in this way can help improve soil fertility and soil health. Biochar can also be used as a sorbent for organic and inorganic contamination of water [16] or for upgrade to biological processes, e.g., methane fermentation [17]. Nevertheless, particular ways of utilization depend on different properties of torrefied biomass/biochars, which profoundly changes with the temperature of a process in the range 200–400 °C [18].

Cultivation of the Oxytree is critical to increasing the biomass yield (Figure 1). Pruning the tree at ~0.05 m above the ground in the middle of May (of the second growing season) is practiced to accelerate the growth and bring out a single straight trunk. During pruning, waste biomass is produced in the amount of 0.11–0.16 Mg·ha^−1^ d.m. (assuming 625 trees per ha) [19]. Still, the amount of this biomass (30% by volume) may be too small for the energy-producing industry. However, after torrefaction, pruned biomass may be a source of additional income for growers in retail.

In our previous data article [19] titled “Fuel Properties of Torrefied Biomass from Pruning of Oxytree,” raw data describing the process of torrefaction and properties of torrefied biomass were presented. The pruned biomass of the Oxytree obtained from the eight different cultivating conditions was torrefied and examined [19]. For this article, biomass data from all eight cultivating conditions were treated as one set because of small differences in their pre-torrefaction properties. This research aimed at the determination of models for the influence of torrefaction temperature and process residence time on the torrefied biomass properties according to the Akaike criterion. Developed models may be used for the determination of the energetic potential of residues from pruning Oxytree and the techno-economic justification of using torrefaction for biomass valorization. In addition, the proposed models could be used to evaluate fuel properties from residues common in logging and horticulture industry because the pruned biomass is similar to common tree branches.

## 2. Materials and Methods

### Models

The schematic diagram of the experiment and data treatment resulting in polynomial model parameter evaluation is shown in Figure 2. The experiment consisted of four elements: (1) Oxytree cultivation and pruning, (2) pruned biomass torrefaction, (3) determination of fuel properties of resulting torrefied biomasses, and (4) estimation of parameters of a polynomial model describing the influence of torrefaction technological parameters (i.e., temperature, residence time) on fuel properties of torrefied biomasses. The details of the experimental methodology of torrefaction process and obtained raw data were presented in the previous data article [19]. The data article contains the results of the pruned biomass process, the fuel properties of raw and torrefied biomass.

The pruned Oxytree biomass was originated from plantations cultivated under 8 different conditions of soil type, irrigation status, and geotextile. Oxytrees were grown on (S) sandy soil (classified as V soil belonging to brunic arenosols) and (C) clay soil (classified as Phaeozems), on which they were irrigated (I+) or not (I−) and had geotextile (G+) or not (G−) (Figure 2). The torrefaction was carried out at temperatures of 200–300 °C with an interval of 20 °C, at residence times 20, 40, and 60 min.

Polynomial models of influence of torrefaction temperature and biomass residence time in the torrefaction reactor on mass and energy efficiency of the torrefaction process, energy densification ratio, organic matter, combustible elements, ash, high heating value, low heating value, and elemental composition of torrefied biomass were built using the raw data (a more detailed description of raw data organized in [19] is presented in Appendix A). The model parameters were estimated due to the non-linear regression analysis. Regression analysis used a 2-degree polynomial with a general form, with intercept (*a*_1_) and 5 regression coefficients (*a*_2–6_) (Equation (1)). The confidence interval of parameter evaluations (*a*_1–6_) was 95%. All results for *p*-value below 0.05 level were assumed to be statistically significant.
(1)$$f\left(T,t\right)=a_{1}+a_{2}\cdot T+a_{3}\cdot T^{2}+a_{4}\cdot t+a_{5}\cdot t^{2}+a_{6}\cdot T\cdot t$$
where:

*f*(*T*, *t*)—the torrefied biomass property obtained under *T*—temperature, and *t*—residence time conditions,

*a*_1_—intercept;

*a*_2–6_—regression coefficient;

*T*—temperature, *T* = 200–300 °C;

*t*—residence time, *t* = 0–60 min.

The standardized regression coefficients (*β*) for each regression coefficients (*a*_2–6_) were determined based on Equation (2). The standardized beta *β* coefficient determines how much (its own) standard deviations will increase or decrease the dependent variable *Y* if the independent variable will be changed by one (its own) standard deviation.
(2)$$\beta =\frac{a_{n}\cdot SD_{Xi}}{SD_{Yi}}$$
where:

*β*—standardized regression coefficient;

*a_n_*—estimated regression coefficient;

*SD_Xi_*—standard deviation of the independent variable x;

*x_i_*—values of subsequent independent variables;

*SD_Yi_*—standard deviation of the dependent variable y;

*y_i_*—values of subsequent dependent variables.

The regression analysis was performed using the Statistica 12 software (StatSoft, Inc., TIBCO Software Inc. Palo Alto, CA, USA). For the determination of model parameters, the degree of matching to raw data, the determination coefficient (R^2^) was calculated. The backward stepwise regression analysis was used for the reduction of insignificant parameters from the model in case of a lack of statistical significance (*p* < 0.05). Then both models were compared with the Akaike Information Criterion (*AIC*) to propose the simplest model with a similar matching to raw data. *AIC* was determined according to the least-squares method (Equation (3)) [20]:(3)$$AIC=n\cdot ln\left(\sum_{i=1}^{n}e_{i}^{2}\right)+2\cdot K$$where:

*AIC*—value of Akaike analysis;

*n*—the number of measurements;

*e*—the value of the rest of the model for particular measurements point;

*K*—number of regression coefficients including intercept (*a_n_*) in model.

Generally, models with a larger number of predictors are more accurate but tend to over-fitting. The over-fitted models are good in predictions of data on which they were built but can result in poorer predictions when other data is used. The *AIC* approach can be used in order to preserve good accuracy and a low number of predictors in compared models. When models for a particular variable are compared, a model with a lower *AIC* is better.

## 3. Results

### Models

Data descriptions. All models are firstly presented by a 3D model figure used to a visualization of data. Next, information about a particular model was summarized in a related table. Each table contains the following information:The first row contains a model to evaluate the particular properties of torrefied biomass and R^2^ value. *AIC* values are also presented in cases where an alternative model (e.g., model 2 or 3) was developed, and the data is presented in Appendix B;The first column shows the intercept *a*_1_ and coefficients *a*_2_–*a*_6_;The second column presents values for particular intercept/coefficients that are used in the model;The third column summarizes standard error calculated for particular intercept/coefficient.The fourth column presents *p*-values (probability value or significance). Statistical significance is assumed when *p* < 0.05).The fifth and sixth columns summarize the lower and upper limit of confidence of intercept/coefficient value.The seventh column summarizes the value of standardized regression coefficients (*β*) for each regression coefficients (*a*_2–6_).

Additional information about data descriptions:
The name (model 1) in table description presents the original model $f\left(T,t\right)=a_{1}+a_{2}\cdot T+a_{3}\cdot T^{2}+a_{4}\cdot t+a_{5}\cdot t^{2}+a_{6}\cdot T\cdot t$. The alternative (model 2) and (model 3) stand for improved versions of a model without insignificant coefficients;Blue lines with circles present in figures stand for raw data used to nonlinear regression;Coefficients with (−) reduce the calculated value of *y* and coefficients with (+) increase the calculated value of *y*. The same system is used for standardized regression coefficients (*β*).

The mass yield (*MY*) of the Oxytree torrefaction is shown in Figure 3 and Table 1. The *MY* decreased as the temperature and process time increased (R^2^ = 0.92). The analysis of data reveals that the increase in temperature is more important to reduce *MY* than the residence time. The mass yield was ~50% for the torrefaction conditions (*T*, *t*) of 300 °C, 60 min. All regression coefficients of the model were statistically significant (*p* < 0.05). In this model, a reduction of *MY* is caused by predictors *T*^2^ (*β* = −1.49), *t* (*β* = −0.19), and *T*∙*t* (*β* = −0.74), respectively.

The torrefaction process led to a reduction in the energy yield (*EY*) of the valorized material (R^2^ = 0.88) (Figure 4). As with *MY*, the predictor T^2^ had the biggest impact on lowering the energy yield value (Table 2). The lowest value of *EY* (<70%) was achieved at 300 °C and 60 min. Each regression coefficient was statistically significant (*p* < 0.05). The EY value is reduced by predictor *T* (*β* = −2.35) and *T*∙*t* (*β* = −1.16). Because the predictor *T*^2^ has the highest negative value, it is reasonable to assume that torrefaction temperature has the biggest impact on decreasing the *EY* (and greater than time).

The increase in energy densification ratio (*EDr*) in the torrefied biomass is one of the main advantages of the biomass torrefaction process. The *EDr* improves with the increase in the process temperature and its duration (Figure 5). The highest energy densification ratio value was ~1.21 for 300 °C and 60 min (Figure 5). The *EDr* model was characterized by a slightly lower R^2^ (0.78) compared with *MY* and *EY*. Regression coefficients of the *EDr* model were statistically significant (*p* < 0.05). The *T*^2^, *t*, and *T*∙*t* predictors cause the increase of *EDr* (*β* = 0.93, 0.78, and 0.23 respectively) (Table 3). The *T* and *t*^2^ predictors were negative (*β* = −0.18 and −0.7, respectively) (Table 3).

The highest organic matter content (*OM*) occurred in the torrefied biomass with the shortest process time and the lowest temperature of 200 °C (Figure 6). The *OM* content in the tested torrefied biomass ranged from 90% to 82%. Because of the large discrepancy in the results (blue vertical lines), the model has R^2^ = 0.61. The regression coefficients of the model describing the organic matter are summarized in Table 4. All regression coefficients of the equation were statistically significant (*p* < 0.05). According to the standardized regression coefficient, predictors *T*, *t*, and *t*^2^ cause an increase of *OM*, whereas predictors *T*^2^ and *T*∙*t* cause a decrease of *OM* (Table 4).

Combustible part (*CP*) had a similar trend to that of *OM.* The *CP* content in the torrefied biomass decrease with time and process temperature (Figure 7). *CP* in the torrefied biomass decreased from 92% to 86% (Figure 7). The content of *CP* was inversely related to ash content (*AC*), i.e., as *CP* decreased*,* the *AC* (Figure 8) increased. The torrefied biomass was characterized by a high *AC* of up to 15%. The *CP* and *AC* models had poor fits (R^2^ = 0.53) because there were significant deviations from mean values of up to 8% (straight blue lines on Figure 7 and Figure 8). The regression coefficients of the *CP* and the *AC* models are presented in Table 5 and Table 6, respectively. All regression coefficients of the equation were statistically significant (*p* < 0.05) for both models. The standardized regression coefficients *β* in the *CP* model had a similar trend as in *OM*; i.e., predictors *T*^2^ and *T*∙*t* cause decrease of *CP* value (Table 5). In the case of *AC* model, predictors *T*, *t*, and *t*^2^ cause the decrease of *AC* value (Table 6).

The high heating value (*HHV*) increased with the process temperature and its duration increase (Figure 9). The highest *HHV =* 23 MJ·kg^−1^ value was recorded at 300 °C and 60 min, whereas the raw biomass had the *HHV* = 18.4 MJ·kg^−1^ [19]. The regression coefficients of the *HHV* model are summarized in Table 7. The a_2_ and a_6_ regression coefficients were not statistically significant (*p* < 0.05) for *HHV* = *a*_1_
*+ a*_2_*·T + a*_3_*·T*^2^
*+ a*_4_*·t + a*_5_*·t*^2^
*+ a*_6_*·T·t* (model 1, Table A1). Consequently, they were removed from the analysis and the estimations were made again for the *HHV = a*_1_
*+ a*_2_*·T*^2^
*+ a*_4_*·t + a*_5_*·t*^2^ (model 2, Figure 9). There were practically no differences between model (1) and (2) according to R^2^ values. However, the Akaike analysis of both models revealed that the model (2) had the *AIC* lower by 2 compared with the model (1). Therefore, the model (2) with a lower value of *AIC* was chosen. According to standardized regression coefficient *β*, the *T*^2^ and t predictors cause an increase of *HHV*, whereas predictor *T*∙*t* a decrease of *HHV* (Table 7).

The lower calorific value (*LHV*) increased with the increase of the process temperature and residence time (Figure 10). The *LHV* was found in torrefied biomass made at 200 °C and 220 °C and ranged from 16 to 20 MJ·kg^−1^, respectively [19]. The highest *LHV* resulted from torrefied biomass generated at 300 °C and 60 min. The regression coefficients of the *LVH* model are presented in Figure 10. Because in the model *LHV = a*_1_
*+ a*_2_*·T + a*_3_*·T*^2^
*+ a*_4_*·t + a*_5_*·t*^2^
*+ a*_6_*·T·t* the *a*_2_ regression coefficient (*p* < 0.05) was not statistically significant (model 1, Table A2), the alternative model *LHV = a*_1_
*+ a*_2_*·T*^2^
*+ a*_3_*·t + a*_4_*·t*^2^
*+ a*_5_*·T·t* was tested (model 2). Again, in this revised model, *a*_5_ was not statistically significant (Table A3), so the model *LHV = a*_1_
*+ a*_2_*·T*^2^
*+ a*_3_*·t + a*_4_*·t*^2^ was tested (model 3). In model 3, all regression coefficients were statistically significant (*p* < 0.05). The R^2^ values were almost the same in each model (~0.82). It can be assumed that the third model, compared to the first and second models, had a better fit because of *AIC* value. The first, second, and third models had an *AIC* of 8137, 8135, and 4904, respectively. For model 3, the predictors of *LHV* (Table 8) have similar trends as predictors of *HHV*.

Figure 11, Figure 12, Figure 13, Figure 14 and Figure 15 present models of the C, H, N, S, O content in the torrefied biomass. The C (Figure 11) and N (Figure 13) contents increased with the increase in process temperature and time. The H (Figure 12) and O (Figure 15) contents had the opposite tends. The 3D model of S content (Figure 14) appears insensitive to temperature or time. Visible changes occurred only above 250 °C and 40 min. In Table 9, Table 10, Table 11, Table 12 and Table 13 a statistical evaluation of these models was presented. For these models, R^2^ ranged from 0.06 to 0.66. The highest R^2^ was for the H model and the lowest for the S model. Relatively high R^2^ = 0.55 was also noted for the O model. Other models had a coefficient of determination <0.5. All regression coefficients were statistically significant (*p* < 0.05) for each model (Table 9, Table 10, Table 11, Table 12 and Table 13). A common finding for C, H, N, S, and O models was the value of standardized regression coefficients β. Predictors related to temperature (*T*, *T*^2^) had a higher absolute value than those related to time (*t*, *t*^2^). However, there was no link where predictors *T* and *T*^2^ had a positive or negative value.

Figure 16 and Figure 17 depict changes in the value of *H:C* and *O:C* ratios depending on the *T* and *t* time. The model of *H:C* ratio (Figure 16) was characterized by high R^2^ (0.82), while the *O:C* ratio model had R^2^ of 0.48. The first model of *H:C* ratio (model 1) had 3 statistically insignificant regression coefficients (*p* < 0.05), R^2^ = 0.82, and *AIC* = 164 (Table A4). Non-significant regression coefficients have been removed, and the second model has been proposed (model 2, Figure 16, Table 14). The second model has the same R^2^ = 0.82 yet with a higher *AIC* = 200 (Figure 16). Elevating the temperature and process time led to a reduction of *H:C* ratio from about 1.5 to 0.9 (%/%) for 300 °C at 20 min and 300 °C at 60 min, respectively (Figure 16). The same change in conditions caused a change in *O:C* ratio from ~0.47 to 0.37 (%/%) (Figure 17). For *H:C* ratio (model 2), the predictor *t*^2^ caused an increase of *H:C* ratio (*β* = 1.68) whereas predictor *T*∙*t* a decrease of *H:C* ratio (*β* = −2.15). With the *O:C* ratio model, the decrease of *O:C* ratio is caused by predictors *T*, *t* and *T*∙*t* for which *β* = −3.03, −0.59, and −0.21 respectively (Table 15).

## 4. Discussion

### 4.1. Models

The mass and energy yield (*MY* and *EY*) of the torrefaction process decreased with increasing *T* and *t*. The *MY* and *EY* values for Oxytree were in the range of other torrefied biomass derived from wood. At temperatures of 275 °C and 300 °C (60 min), the *MY* value for spruce was ~70% and ~50% [21], i.e., corresponding to those reported in Figure 1. At temperatures of 275 °C and 300 °C (60 min), the *MY* value for spruce was ~70% and ~50% [16], i.e., corresponding to those reported in Figure 1. For willow torrefied in 15 min at 250 °C, the *MY* was 70% [22]. It is much less than for torrefied pruned biomass of Oxytree for which *MY* at 250 °C, 20 min was 90% (Figure 1). The *MY* and *EY* in the case of torrefied willow at a process temperature of 230–290 °C was 95%–72%, and *EY* was 97%–79% [23]. In the case of Oxytree, these values ranged from 90%–50% and 90%–70%, respectively (Figure 3 and Figure 4). Energy efficiency for spruce torrefaction at 225–300 °C was between 93%–68%, and the *EDr* at 300 °C was 1.2 [24], i.e., the same as for the pruned torrefied Oxytree biomass (Figure 5).

The models for *MY*, *EY*, and *EDr* were characterized by a high determination coefficient R^2^ of 0.78–0.92, which means that the proposed models can be considered suitable for describing the torrefaction of pruned biomass from a cultivation treatment.

The content of *OM* in the torrefied Oxytree ranged from 90% to 80%. The model describing the value of *OM* had a relatively low R^2^ (0.63). The decrease in the *OM* resulted from the decomposition of organic compounds under the influence of T and their degassing (torgas). The lower fit of the model to the data was likely due to the high variability of empirical data around the average (illustrated with the blue vertical lines). Nevertheless, it showed accurately the trends of the *OM* loss along with the increase of temperature and time.

Similar low coefficients of determination were obtained for the *CP* and *AC* (R^2^ = 0.53). This was similar, as in the case of *OM*, because of the large variation in measurement data. *CP* is associated with the ash content. As the *CP* decreases, the *AC* increases. Torrefaction causes a decrease in *CP* and an increase in *AC*. It is associated with the degassing of combustibles that are released during the torrefaction process. The *AC* in the torrefied pruned Oxytree biomass ranged from 7% to 1% [19]. This is a much higher value than that found in torrefied wood from torrefied pine 0.15%–0.21% [25] and birch 0.23%–0.38% [26]. The AC values obtained for Oxytree were closer to the corn stover 10%–12% [27]. The model of *HHV = a*_1_
*+ a*_2_*·T*^2^
*+ a*_4_*·t + a*_5_*·t*^2^ form was better than *HHV = a*_1_
*+ a*_2_*·T + a*_3_*·T*^2^
*+ a*_4_*·t + a*_5_*·t*^2^
*+ a*_6_*·T·t* because it had a lower number of parameters, so it was easier to use. In addition, the model had a lower *AIC* value. The differences were insignificant, but the shorter model had an *AIC* of 8126 compared to 8128, whereas the R^2^ was almost the same for both.

The *LHV* of the torrefied pruned Oxytree biomass at 270 °C and 30 min was over 21.5 MJ·kg^−1^ (Figure 9) and was comparable with the calorific value of willow torrefied in the same conditions [28]. The calorific value of the torrefaction from the pruned Oxytree biomass generated at 265 °C and 60 min was above 20 MJ·kg^−1^ (Figure 10). This value was ~1–3 MJ·kg^−1^ lower than in the case of torrefied eucalyptus, poplar, and pine at 265 °C and 105 min [29].

Models describing the change in the content of elements suggest that increasing the torrefaction temperature leads to an increase in the C content and a decrease in the content of O and H. An interesting tendency can be observed in the N model, i.e., the amount of N increased with increasing T and t. This finding is opposite to the typical N content [30,31]. The S model had a low R^2^ = 0.06. According to the S model (Figure 14), the S content decreases from 260 °C, 20 min, and 200 °C, 40 min, nevertheless, the decrease was small. The smallest S content was obtained at 300 °C where S was ~0.19%. In this study, a trend of S changes was not clear because of variability in measurements. Ren et al. [32] reported 30%–80% S loss during torrefaction of herbaceous, crop, and woody biomass, depending on feedstock. Lack of greater decrease of S content in this research was probably caused by the decrease of other more reactive elements. During the torrefaction process, the decrease in O and H content was mainly due to the weakness of bonded structures such as -OH [33]. Similarly, a drop in N content should be observed due to losses of weakly bonded structures, e.g., -NH_2_ [33]. The increase in the C content resulted likely from slower decomposition in comparison with other elements [34].

The decrease of *O:C* with the *T* rise might be also attributed to the loss of hydrophilic surfaces [35]. In addition, the C losses were smaller than in the case of O. The decrease in *H:C* resulted, similarly to the decrease in the content of *O:C,* from the faster decomposition of substances containing H in relation to C. The lower values of *O:C* or *H:C* ratios cause the higher energy content of fuel feedstock [36]. Torrefied pruned Oxytree biomass produced at 300 °C and 60 min was characterized by the lowest values of *O:C* and *H:C* of 0.9 and 0.375 (%/%), respectively. *O:C* and *H:C* ratios for wood biomass were >1.4 and >0.65, respectively [36]. For bituminous coal *O:C* was 1.2 (%/%)and *H:C* was 0.125 (%/%) [37]. The values for the torrefied pruned Oxytree biomass were similar to the value of *Gmelina arborea*, torrefied in the same conditions (300 °C, 60 min) [36].

The standardized regression coefficients β are challenging to interpret because predictors are correlated in each other. For example, when one predictor related to *T* has a positive impact on the dependent value, the second predictors (e.g., *T*^2^, or *T*∙*t*) not necessarily do. In almost all cases, the correlated predictors had an opposing impact. One common characteristic which could be observed based on *β* in most of the presented models is that predictors which depended on temperature had an absolute impact greater than these ones related to the time. Based on this, it can be assumed that temperature has a greater impact on the properties of torrefied pruned biomass of Oxytree.

### 4.2. Evaluations of the Value of Torrefied Residue Biomass

The common assumption is that the amount of biomass produced during the pruning treatment of the Oxytree plantation is too small to be economically used for energy purposes. Nevertheless, assuming that the material tested in [19] has properties similar to branches (and others residues) that make up waste at Oxytree harvesting (up to 30% of the weight of the tree), a simple model is proposed here for calculating the value of torrefied biomass produced in relation to commercial coal fuel available on the market, depending on the T and the duration of the torrefaction process (Figure 18). The model also theoretically calculates the maximum profit from the waste mass on the plantation. The calculations assume that part of the terrified biomass is used to maintain the torrefaction process. Calculations do not include labor costs, harvesting, transport, processing, and other costs related to the torrefaction process as well as the distribution of produced fuel.

Data for calculations:mass of Oxytree residues, Mg; assumed 1 Mg;the moisture content of Oxytree residues, %; assumed 50%;torrefaction parameters temperature and time; assumed to be 200 °C and 20 min.

#### 4.2.1. Initial Calculations

The dry mass of Oxytree residues:(4)$$mr_{d}=mr_{w}-mr_{w}\cdot MC$$where:

*mr_d_*—dry mass of Oxytree residues, M_g_,

*mr_w_*—wet mass of Oxytree residues, M_g_,

*MC*—moisture content of Oxytree residues, %.

Amount of water in Oxytree residues:(5)$$m_{w}=mr_{w}-mr_{d}$$where:

*m_w_*—mass of water in Oxytree residues, Mg.

#### 4.2.2. Main Properties of Torrefied Biomass Calculations

Mass yield of torrefaction based on Figure 3
(6)$$MY=\;0.891816+0.003525\cdot T-0.000013\cdot T^{2}-0.001684\cdot t+0.000062\cdot t^{2}-0.000025\cdot T\cdot t$$
where:

*MY*—mass yield of torrefaction process, %;

*T*—temperature of torrefaction, °C;

*t*—time of torrefaction, min.

Mass of torrefied biomass after torrefaction
(7)$$mtb=mr_{d}\cdot MY$$
where:

*mtb*—mass of torrefied biomass after torrefaction process at *T*, and *t* conditions.

*LHV* of torrefied biomass based on Figure 10
(8)$$LHV_{tb}=\left(12394.39+0.07\cdot T^{2}+90.68\cdot t-0.79\cdot t^{2}\right)/1000$$
where:

*LHV_tb_*—the low heating value of torrefied biomass depending of torrefaction conditions, MJ·kg^−1^;

1000—conversion of kJ to MJ.

The total energy in torrefied biomass
(9)$$E_{tb}=MY\cdot LHV\cdot 1000$$
where:

*E_tb_*—energy in torrefied biomass, kJ;

1000—conversion of Mg to kg.

#### 4.2.3. Energy Need to Torrefaction Process

Data for calculations [38]:*T_a_*—ambient temperature, °C, assumed 15 °C;*T_b_*—boiling point of water, 100 °C;latent heat of water vaporization, 2500 kJ·kg^−1^ [39] ;specific heat of water, 4.18 kJ·kg^−1^ [39];specific heat of wood, kJ·kg^−1^, assumed 1.6 kJ·kg^−1^ [40].

The energy needed to heat water contained in Oxytree residues
(10)$$E_{w}=m_{w}\cdot Cp_{water}\cdot \left(T_{b}-T_{a}\right)$$
where:

*Ew*—energy needed to heat water contained in Oxytree residues, MJ;

*Cp_water_*—specific heat of water, 4.18 kJ·kg^−1^.

The energy needed to water vaporization
(11)$$E_{ev}=m_{w}\cdot L_{h}$$
where:

*E_ev_*—energy needed to vaporization of water contained in Oxytree residues, MJ;

*L_h_*—latent heat of water vaporization, kJ·kg^−1^.

The energy needed to heat Oxytree residues during torrefaction
(12)$$E_{hw}=mr_{d}\cdot Cp_{wood}\cdot \left(T-T_{a}\right)$$
where:

*E_hw_*—energy needed to heat Oxytree residues from ambient to torrefaction temperature, MJ;

*Cp_wood_*—specific heat of wood, kJ·kg^−1^.

Total energy needed to torrefied Oxytree residues
(13)$$E=E_{w}+E_{ev}+E_{hw}\;$$
where:

E—energy needed to torrefied Oxytree residues

#### 4.2.4. Estimation of the Value of Torrefied Biomass

Estimation was done based on the price of commercial coal fuel available in Poland’s market in 2019 and its *LHV*. The value in PLN has been converted to € at the current exchange rate.

Data for calculations:Price of commercial coal fuel, €·Mg^−1^, assumed 170 €·Mg^−1^ [41];*LHV* of commercial coal fuel, MJ·kg^−1^, assumed 23 MJ·kg^−1^ [41].

The estimated value of torrefied biomass
(14)$$V_{tb}=\frac{V_{ccf}\cdot LHV_{tb}}{LHV_{ccf}}$$
where:

*V_tb_*—the estimated value of torrefied biomass, €·Mg^−1^;

*Vccf*—value (price) of commercial coal fuel, €·Mg^−1^;

*LHV_ccf_*—low heating value of commercial coal fuel, MJ·kg^−1^.

#### 4.2.5. Profit from Torrefied Oxytree Residues

Mass of torrefied Oxytree residues net (when assumed that part of it is used as fuel to the process of torrefaction)
(15)$$mtb_{n}=\frac{E_{tb}-E}{LHV_{tb}}$$
where:

*mtb_n_*—a mass of torrefied Oxytree residues net, Mg.

The evaluations of the value of torrefied biomass were completed for 1 Mg of Oxytree wet residues. The moisture content in Oxytree residues was assumed as 50%. Based on Solver a Microsoft Excel add-in program, the best conditions for the torrefaction process (in terms of economics) were *T* = 200 °C, and *t* = 20 min. For these conditions, *MY* of torrefied biomass was 97% and *LHV* of torrefied biomass was 16.7 MJ·kg^−1^. In these conditions, the calculated value of produced torrefied biomass was 123.38 €·Mg^−1^ d.m., while the net mass obtained after torrefaction was 0.39 Mg d.m. The evaluated value of torrefied biomass from 1 Mg of Oxytree wet residues (containing 50% of moisture) was €44.92.

The presented simple model of evaluation of the value of torrefied Oxytree residues as a fuel is the first step to the evaluation of the profitability of utilization of torrefaction technology to Oxytree residues. The model has been based on simple assumptions, and thus, it cannot be used as a fully-fledged tool to evaluate the economic value of torrefied biomass yet. For prices of fuel such as coal, the impact has many factors, such as ash content, grindability, fraction, etc. Nevertheless, after knowing these factors and their impact on the price of coal, the presented model could be extended and improved. The situation is the same with the estimations of the cost of production of torrefied biomass. A more complete and improved analysis of all types of costs is warranted.

## 5. Conclusions

In this article, we used raw data of torrefied pruned biomass of Oxytree and developed mathematical models describing torrefied pruned Oxytree properties. Presented models of mass yield, energy yield, energy densification ratio, *HHV*, and *LHV* are characterized by R^2^ > 0.78. Thus, the newly developed models could be used for describing the process of torrefaction of biomass originating from Oxytree pruning. The other models still describe the process trends well albeit with large standard deviations in the measurement data. The energetic properties of torrefied Oxytree biomass are comparable to other woody biomass. The highest *HHV* of torrefied biomass was 21 MJ·kg^−1^ at 300 °C and 20 min. The study found that the most beneficial economic aspect parameters of torrefaction are 200 °C and 20 min. These parameters provide the greatest profit and the smallest energetic losses.

## Figures and Tables

**Figure 1 materials-12-02228-f001:**
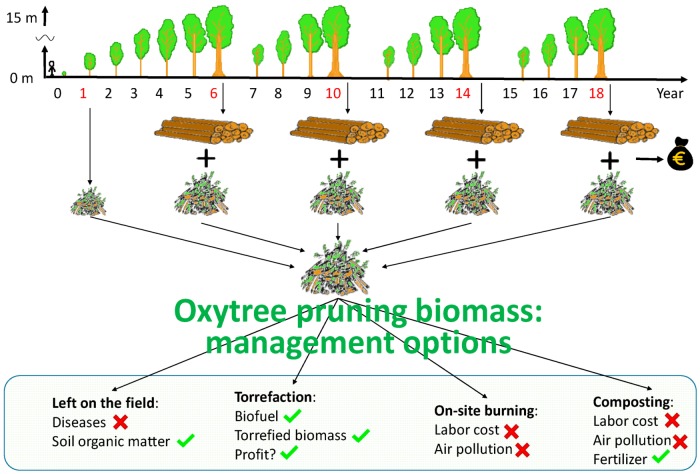
Graphic presentation of the current and proposed utilization of biomass residues on a plantation.

**Figure 2 materials-12-02228-f002:**
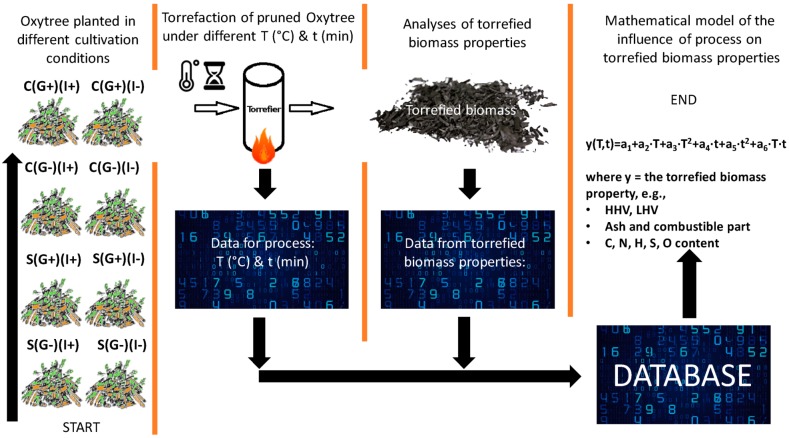
Diagram of experimental design and data evaluation.

**Figure 3 materials-12-02228-f003:**
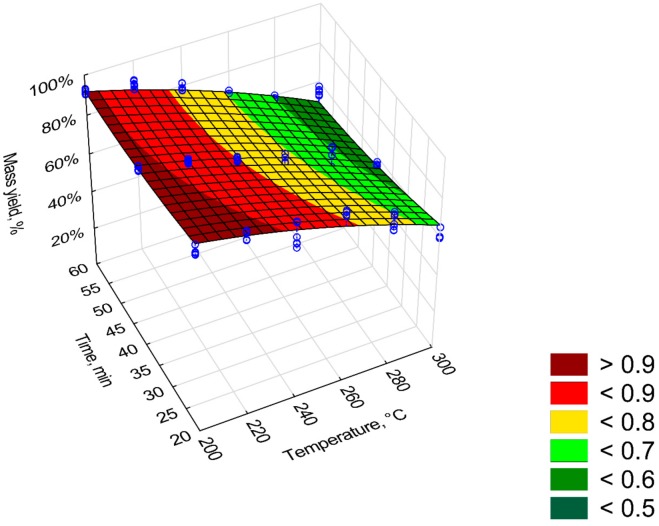
Visualization of 3D model of mass yield (*MY*) of pruned biomass torrefaction.

**Figure 4 materials-12-02228-f004:**
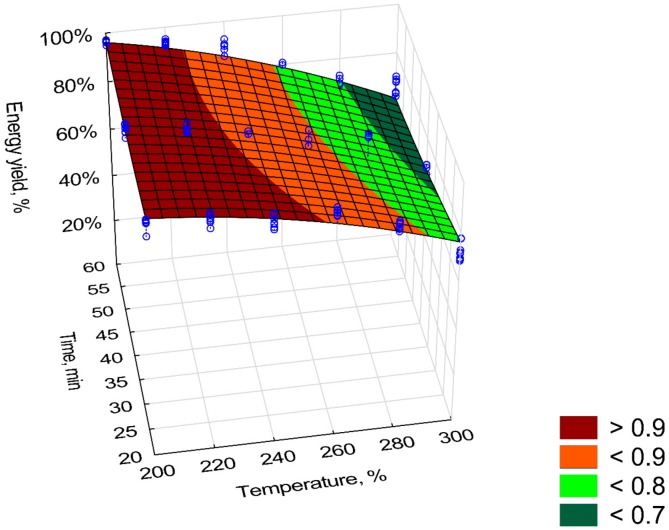
Visualization of 3D model of energy yield (*EY*) of pruned biomass torrefaction.

**Figure 5 materials-12-02228-f005:**
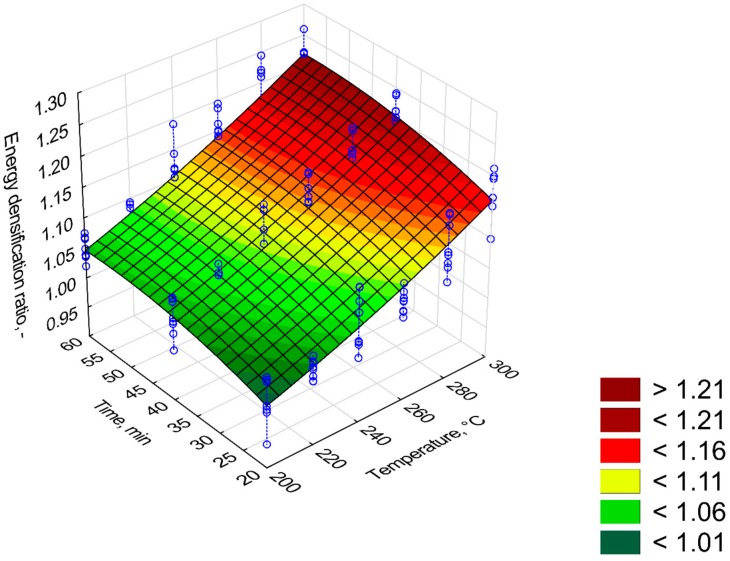
Visualization of 3D model of energy densification ratio (*EDr*) of pruned biomass torrefaction.

**Figure 6 materials-12-02228-f006:**
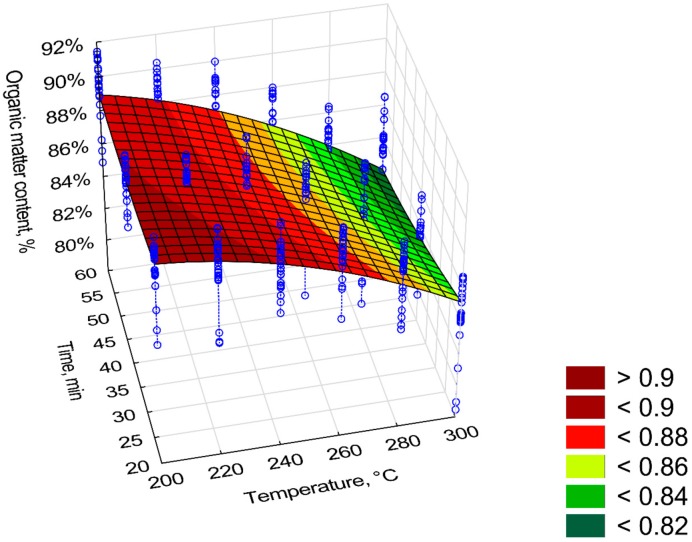
Visualization of 3D model of organic matter (*OM*) content of torrefied pruned biomass.

**Figure 7 materials-12-02228-f007:**
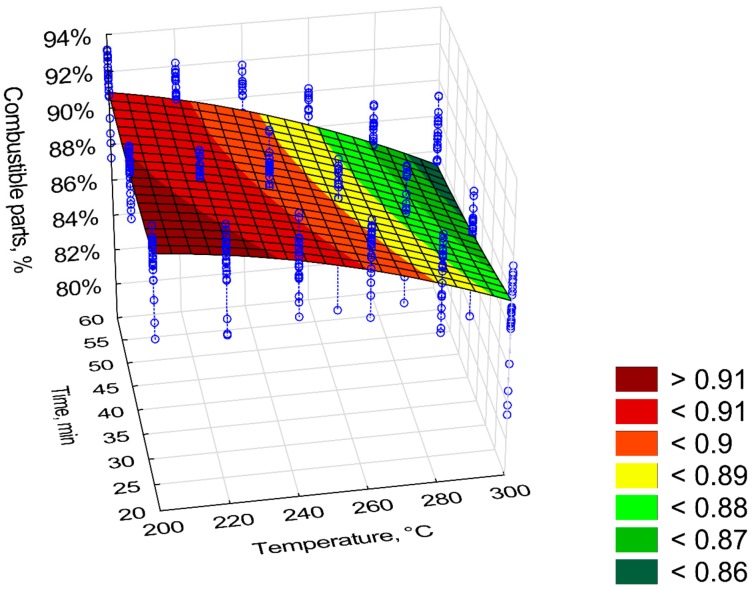
Visualization of 3D model of the combustible part (*CP*) of torrefied pruned biomass.

**Figure 8 materials-12-02228-f008:**
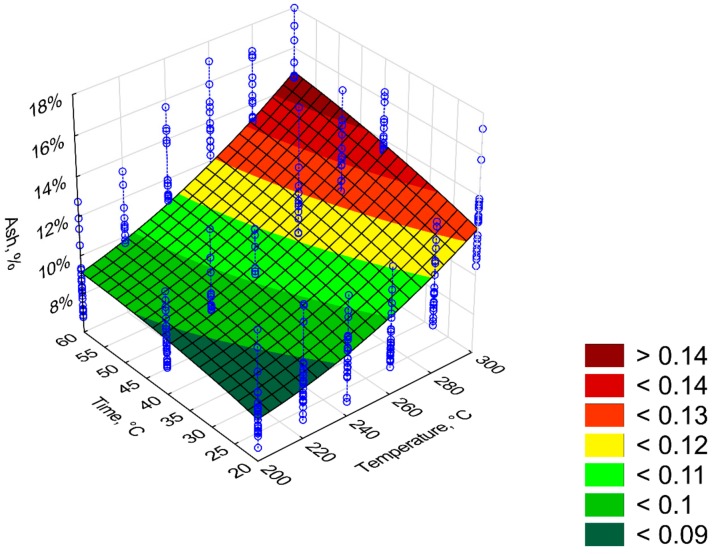
Visualization of 3D model of ash content (*AC*) of torrefied pruned biomass.

**Figure 9 materials-12-02228-f009:**
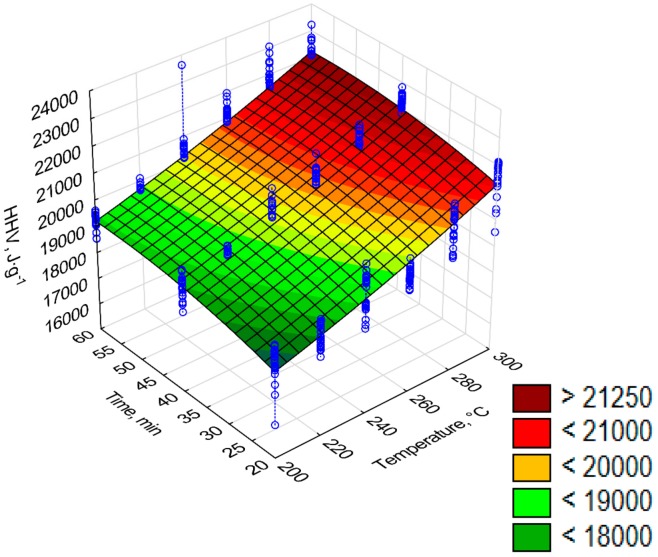
Visualization of 3D model of *HHV* of torrefied pruned biomass (model 2).

**Figure 10 materials-12-02228-f010:**
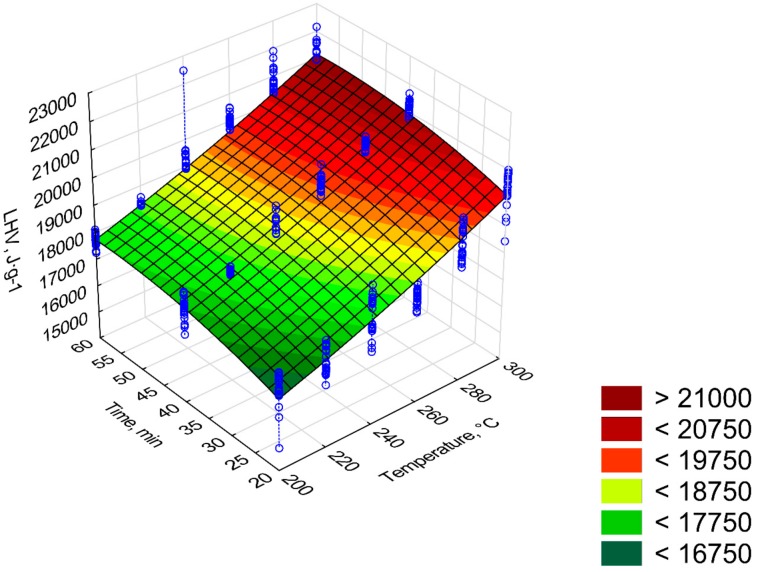
Visualization of 3D model of *LHV* of torrefied pruned biomass (model 3).

**Figure 11 materials-12-02228-f011:**
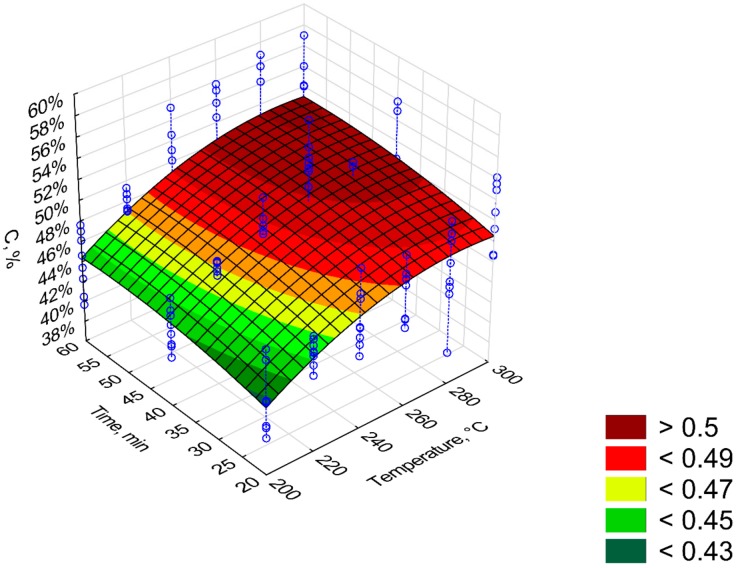
Visualization of 3D model of C content of torrefied pruned biomass.

**Figure 12 materials-12-02228-f012:**
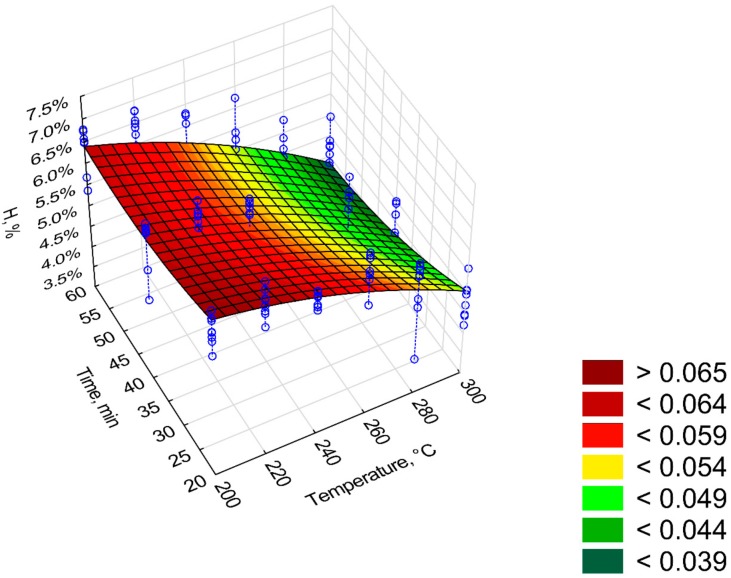
Visualization of 3D model of H content of torrefied pruned biomass.

**Figure 13 materials-12-02228-f013:**
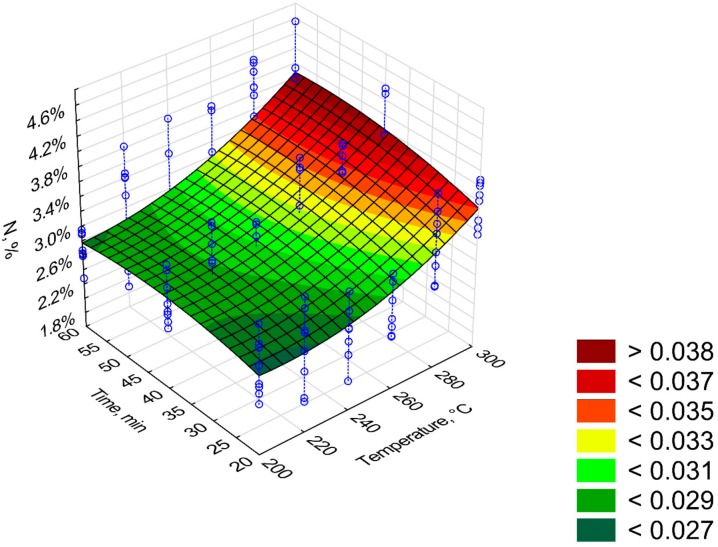
Visualization of 3D model of N content of torrefied pruned biomass.

**Figure 14 materials-12-02228-f014:**
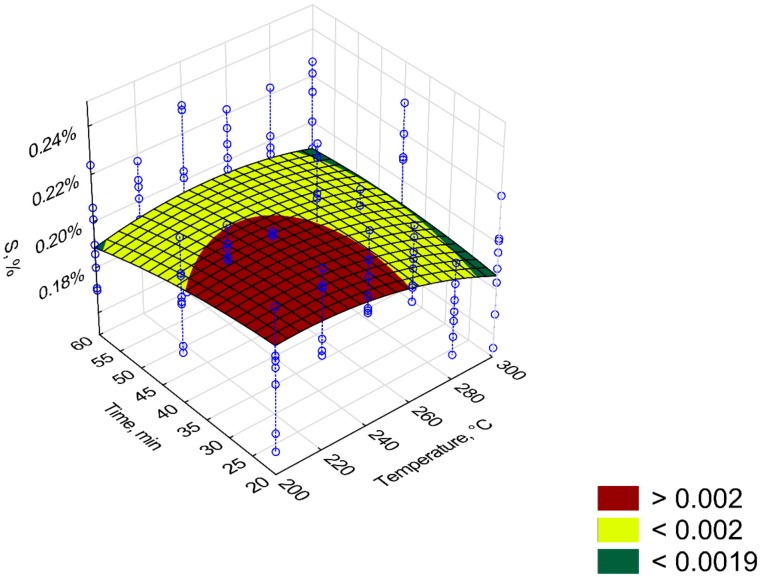
Visualization of 3D model of S content of torrefied pruned biomass.

**Figure 15 materials-12-02228-f015:**
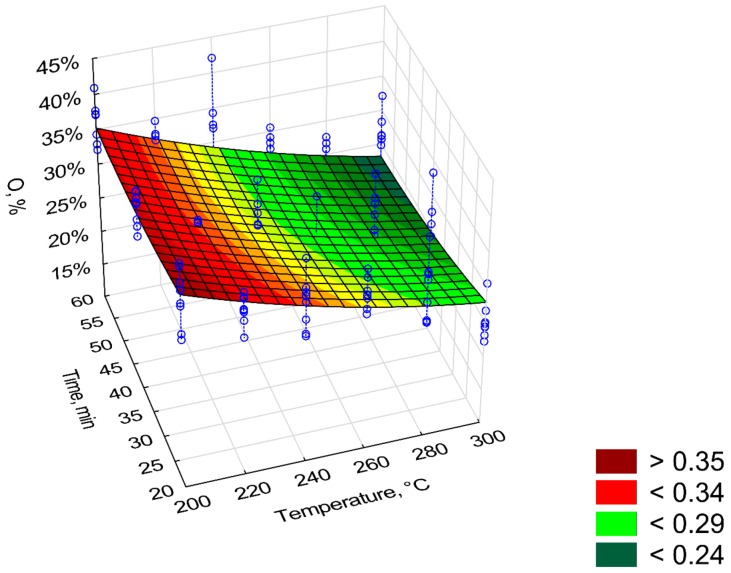
Visualization of 3D model of O content of torrefied pruned biomass.

**Figure 16 materials-12-02228-f016:**
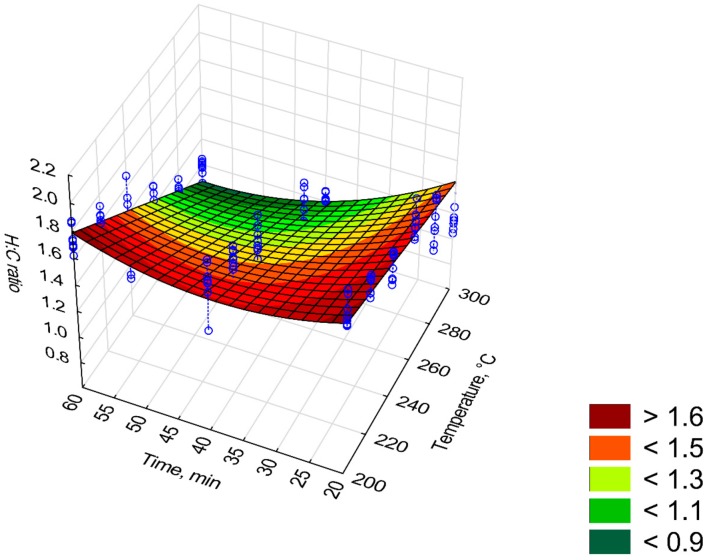
Visualization of 3D model of *H:C* ratio of torrefied pruned biomass (model 2).

**Figure 17 materials-12-02228-f017:**
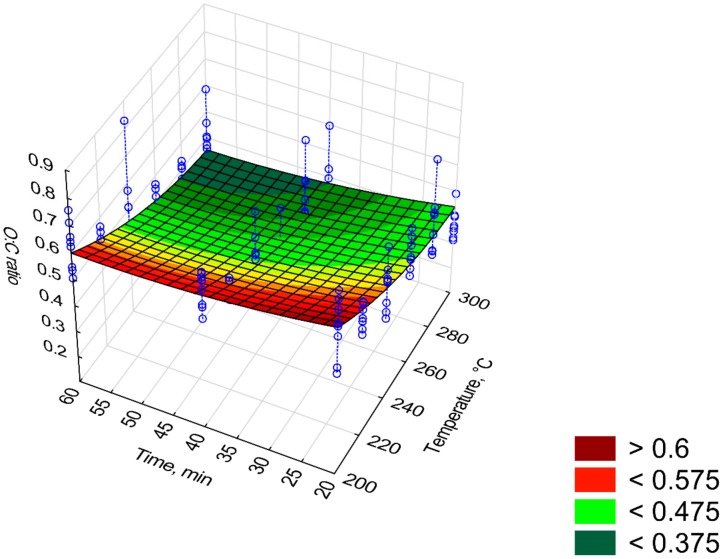
Visualization of 3D model of *O:C* ratio of torrefied pruned biomass.

**Figure 18 materials-12-02228-f018:**
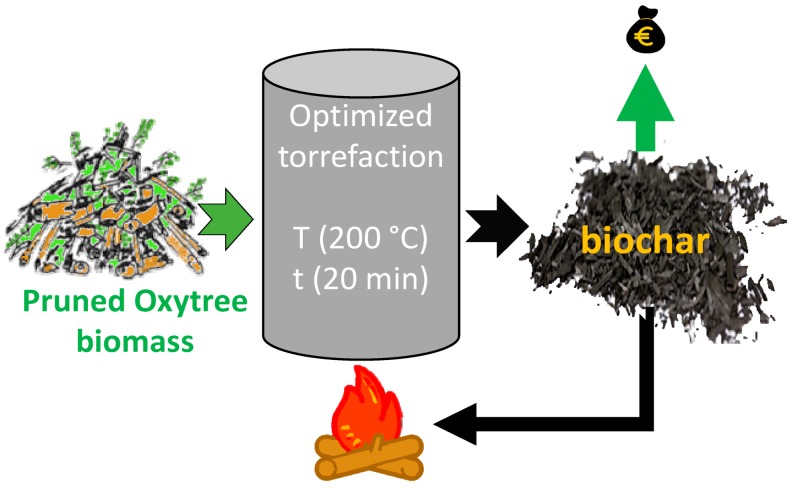
Graphic presentation of the benefits of the pruned Oxytree torrefaction concept.

**Table 1 materials-12-02228-t001:** Statistical evaluation of mass yield (*MY*) model of pruned biomass torrefaction.

Intercept/Coefficient	Value of Intercept/Coefficient	Standard Error	*p*	Lower Limit of Confidence	Upper Limit of Confidence	Standardized *β* Coefficient
*a* _1_	0.891816	0.223378	0.000000	0.450129	1.333503	–
*a* _2_	0.003525	0.001746	0.000000	0.000074	0.006977	0.83
*a* _3_	−0.000013	0.000000	0.000000	−0.000013	−0.000013	−1.49
*a* _4_	−0.001684	0.002135	0.000000	−0.005905	0.002538	−0.19
*a* _5_	0.000062	0.000018	0.000000	0.000025	0.000098	0.56
*a* _6_	−0.000025	0.000000	0.000000	−0.000025	−0.000025	−0.74

*MY* = *a*_1_ + *a*_2_·*T* + *a*_3_·*T*^2^ + *a*_4_·*t* + *a*_5_·*t*^2^ + *a*_6_·*T*·*t*, R^2^ = 0.92, (model 1).

**Table 2 materials-12-02228-t002:** Statistical evaluation of energy yield (*EY*) model of pruned biomass torrefaction.

Intercept/Coefficient	Value of Intercept/Coefficient	Standard Error	*p*	Lower Limit of Confidence	Upper Limit of Confidence	Standardized *β* Coefficient
*a* _1_	0.429884	0.219439	0.000000	−0.004012	0.863781	–
*a* _2_	0.006285	0.001715	0.000000	0.002894	0.009675	1.85
*a* _3_	−0.000016	0.000000	0.000000	−0.000016	−0.000016	−2.35
*a* _4_	0.002472	0.002097	0.000000	−0.001675	0.006619	0.35
*a* _5_	0.000037	0.000018	0.000000	0.000002	0.000073	0.42
*a* _6_	−0.000031	0.000000	0.000000	−0.000031	−0.000031	−1.16

*EY* = *a*_1_ + *a*_2_·*T* + *a*_3_·*T*^2^ + *a*_4_·*t* + *a*_5_·*t*^2^ + *a*_6_·*T*·*t*, R^2^ = 0.88, (model 1).

**Table 3 materials-12-02228-t003:** Statistical evaluation of energy densification ratio (*EDr*) model of pruned biomass torrefaction.

Intercept/Coefficient	Value of Intercept/Coefficient	Standard Error	*p*	Lower Limit of Confidence	Upper Limit of Confidence	Standardized *β* Coefficient
*a* _1_	0.860189	0.182285	0.000000	0.499756	1.220621	–
*a* _2_	−0.000366	0.001424	0.000000	−0.003183	0.002450	−0.18
*a* _3_	0.000004	0.000000	0.000000	0.000004	0.000004	0.93
*a* _4_	0.003294	0.001742	0.000000	−0.000151	0.006739	0.78
*a* _5_	−0.000037	0.000015	0.000000	−0.000066	−0.000007	−0.70
*a* _6_	0.000004	0.000000	0.000000	0.000004	0.000004	0.23

*EDr* = *a*_1_ + *a*_2_·*T* + *a*_3_·*T*^2^ + *a*_4_·*t* + *a*_5_·*t*^2^ + *a*_6_·*T*·*t*, R^2^ = 0.78, (model 1).

**Table 4 materials-12-02228-t004:** Statistical evaluation of organic matter (*OM*) content model torrefied of pruned biomass.

Intercept/Coefficient	Value of Intercept/Coefficient	Standard Error	*p*	Lower Limit of Confidence	Upper Limit of Confidence	Standardized *β* Coefficient
*a* _1_	0.764595	0.059289	0.000000	0.648061	0.881130	–
*a* _2_	0.001510	0.000463	0.000000	0.000600	0.002421	1.76
*a* _3_	−0.000004	0.000000	0.000000	−0.000004	−0.000004	−2.21
*a* _4_	0.000138	0.000567	0.000000	−0.000976	0.001252	0.08
*a* _5_	0.000008	0.000000	0.000000	0.000008	0.000008	0.35
*a* _6_	−0.000005	0.000000	0.000000	−0.000005	−0.000005	−0.76

*OM* = *a*_1_ + *a*_2_·*T* + *a*_3_·*T*^2^ + *a*_4_·*t* + *a*_5_·*t*^2^ + *a*_6_·*T*·*t*, R^2^ = 0.61, (model 1).

**Table 5 materials-12-02228-t005:** Statistical evaluation of the combustible part (*CP*) model of torrefied pruned biomass.

Intercept/Coefficient	Value of Intercept/Coefficient	Standard Error	*p*	Lower Limit of Confidence	Upper Limit of Confidence	Standardized *β* Coefficient
*a* _1_	0.838668	0.054980	0.000000	0.730603	0.946733	–
*a* _2_	0.000997	0.000430	0.000000	0.000152	0.001841	1.33
*a* _3_	−0.000003	0.000000	0.000000	−0.000003	−0.000003	−1.81
*a* _4_	0.000029	0.000526	0.000000	−0.001004	0.001062	0.02
*a* _5_	0.000005	0.000000	0.000000	0.000005	0.000005	0.28
*a* _6_	−0.000004	0.000000	0.000000	−0.000004	−0.000004	−0.60

*CP* = *a*_1_ + *a*_2_·*T* + *a*_3_·*T*^2^ + *a*_4_·*t* + *a*_5_·*t*^2^ + *a*_6_·*T*·*t*, R^2^ = 0.53, (model 1).

**Table 6 materials-12-02228-t006:** Statistical evaluation of ash content (*AC*) model of torrefied pruned biomass.

Intercept/Coefficient	Value of Intercept/Coefficient	Standard Error	*p*	Lower Limit of Confidence	Upper Limit of Confidence	Standardized *β* Coefficient
*a* _1_	0.161333	0.054979	0.000000	0.053268	0.269398	–
*a* _2_	−0.000997	0.000430	0.000000	−0.001841	−0.000152	−1.33
*a* _3_	0.000003	0.000000	0.000000	0.000003	0.000003	1.81
*a* _4_	−0.000029	0.000525	0.000000	−0.001062	0.001004	−0.02
*a* _5_	−0.000005	0.000000	0.000000	−0.000005	−0.000005	−0.28
*a* _6_	0.000004	0.000000	0.000000	0.000004	0.000004	0.60

Ash (*AC*) = *a*_1_ + *a*_2_·*T* + *a*_3_·*T*^2^ + *a*_4_·*t* + *a*_5_·*t*^2^ + *a*_6_·*T*·*t*, R^2^ = 0.53, (model 1).

**Table 7 materials-12-02228-t007:** Statistical evaluation of *HHV* model of torrefied pruned biomass (model 2).

Intercept/Coefficient	Value of Intercept/Coefficient	Standard Error	*p*	Lower Limit of Confidence	Upper Limit of Confidence	Standardized *β* Coefficient
*a* _1_	14,572.93	235.6392	0.000000	14109.78	15,036.09	–
*a* _2_	0.06	0.0016	0.000000	0.06	0.06	0.83
*a* _3_	76.79	12.0003	0.000000	53.20	100.38	1.00
*a* _4_	−0.67	0.1485	0.000009	−0.96	−0.38	−0.70

*HHV* = *a*_1_ + *a*_2_·*T*^2^ + *a*_3_·*t* + *a*_4_·*t*^2^, R^2^ = 0.79, (model 2), *AIC* = 8126.

**Table 8 materials-12-02228-t008:** Statistical evaluation of *LHV* model of torrefied pruned biomass (model 3).

Intercept/Coefficient	Value of Intercept/Coefficient	Standard Error	*p*	Lower Limit of Confidence	Upper Limit of Confidence	Standardized *β* Coefficient
*a* _1_	12,394.39	238.9256	0.000000	11,924.77	12,864.00	–
*a* _2_	0.07	0.0017	0.000000	0.07	0.07	0.84
*a* _3_	90.68	12.1676	0.000000	66.76	114.59	1.06
*a* _4_	−0.79	0.1505	0.000000	−1.09	−0.50	−0.75

*LHV* = *a*_1_ + *a*_2_·*T*^2^ + *a*_3_·*t* + *a*_4_·*t*^2^, R^2^ = 0.82, (model 3), *AIC* = 4904.

**Table 9 materials-12-02228-t009:** Statistical evaluation of C content model of torrefied pruned biomass.

Intercept/Coefficient	Value of Intercept/Coefficient	Standard Error	*p*	Lower Limit of Confidence	Upper Limit of Confidence	Standardized *β* Coefficient
*a* _1_	−0.212855	0.188451	0.000000	−0.585480	0.159770	−
*a* _2_	0.004700	0.001473	0.000000	0.001788	0.007612	3.80
*a* _3_	−0.000008	0.000000	0.000000	−0.000008	−0.000008	−3.35
*a* _4_	0.001477	0.001801	0.000000	−0.002084	0.005039	0.57
*a* _5_	−0.000017	0.000015	0.000000	−0.000047	0.000014	−0.52
*a* _6_	0.000002	0.000000	0.000000	0.000002	0.000002	0.16

C = *a*_1_ + *a*_2_·*T* + *a*_3_·*T*^2^ + *a*_4_·*t* + *a*_5_·*t*^2^ + *a*_6_·*T*·*t*, R^2^ = 0.35, (model 1).

**Table 10 materials-12-02228-t010:** Statistical evaluation of H content model of torrefied pruned biomass.

Intercept/Coefficient	Value of Intercept/Coefficient	Standard Error	*p*	Lower Limit of Confidence	Upper Limit of Confidence	Standardized *β* Coefficient
*a* _1_	0.042265	0.032430	0.000000	−0.021858	0.106388	−
*a* _2_	0.000377	0.000253	0.000000	−0.000124	0.000878	1.27
*a* _3_	−0.000001	0.000000	0.000000	−0.000001	−0.000001	−1.74
*a* _4_	−0.000164	0.000310	0.000000	−0.000777	0.000449	−0.27
*a* _5_	0.000006	0.000000	0.000000	0.000006	0.000006	0.75
*a* _6_	−0.000002	0.000000	0.000000	−0.000002	−0.000002	−0.85

H = *a*_1_ + *a*_2_·*T* + *a*_3_·*T*^2^ + *a*_4_·*t* + *a*_5_·*t*^2^ + *a*_6_·*T*·*t*, R^2^ = 0.66, (model 1).

**Table 11 materials-12-02228-t011:** Statistical evaluation of N content model of torrefied pruned biomass.

Intercept/Coefficient	Value of Intercept/Coefficient	Standard Error	*p*	Lower Limit of Confidence	Upper Limit of Confidence	Standardized *β* Coefficient
*a* _1_	0.068473	0.023985	0.000000	0.021047	0.115899	−
*a* _2_	−0.000429	0.000187	0.000000	−0.000800	−0.000059	−2.59
*a* _3_	0.000001	0.000000	0.000000	0.000001	0.000001	3.01
*a* _4_	0.000134	0.000229	0.000000	−0.000319	0.000588	0.39
*a* _5_	−0.000003	0.000000	0.000000	−0.000003	−0.000003	−0.71
*a* _6_	0.000001	0.000000	0.000000	0.000001	0.000001	0.51

N = *a*_1_ + *a*_2_·*T* + *a*_3_·*T*^2^ + *a*_4_·*t* + *a*_5_·*t*^2^ + *a*_6_·*T*·*t*, R^2^ = 0.41, (model 1).

**Table 12 materials-12-02228-t012:** Statistical evaluation of S content model of torrefied pruned biomass.

Intercept/Coefficient	Value of Intercept/Coefficient	Standard Error	*p*	Lower Limit of Confidence	Upper Limit of Confidence	Standardized *β* Coefficient
*a* _1_	0.001055	0.001325	0.000000	−0.001566	0.003676	−
*a* _2_	0.000010	0.000010	0.000000	−0.000010	0.000031	1.41
*a* _3_	2.61 × 10^−8^	0.000000	0.000000	0.000000	0.000000	−1.81
*a* _4_	−0.000008	0.000013	0.000000	−0.000033	0.000017	−0.52
*a* _5_	6.77 × 10^−8^	0.000000	0.000000	0.000000	0.000000	−0.36
*a* _6_	4.63 × 10^−8^	0.000000	0.000000	0.000000	0.000000	0.81

S = *a*_1_ + *a*_2_·*T* + *a*_3_·*T*^2^ + *a*_4_·*t* + *a*_5_·*t*^2^ + *a*_6_·*T*·*t*, R^2^ = 0.06, (model 1).

**Table 13 materials-12-02228-t013:** Statistical evaluation of O content model of torrefied pruned biomass.

Intercept/Coefficient	Value of Intercept/Coefficient	Standard Error	*p*	Lower Limit of Confidence	Upper Limit of Confidence	Standardized *β* Coefficient
*a* _1_	0.873544	0.207078	0.000000	0.464088	1.283001	−
*a* _2_	−0.003199	0.001618	0.000000	−0.006398	0.000001	−1.96
*a* _3_	0.000005	0.000000	0.000000	0.000005	0.000005	1.42
*a* _4_	−0.001367	0.001979	0.000000	−0.005280	0.002547	−0.40
*a* _5_	0.000022	0.000017	0.000000	−0.000011	0.000055	0.52
*a* _6_	−0.000005	0.000000	0.000000	−0.000005	−0.000005	−0.41

O = *a*_1_ + *a*_2_·*T* + *a*_3_·*T*^2^ + *a*_4_·*t* + *a*_5_·*t*^2^ + *a*_6_·*T*·*t*, R^2^ = 0.55, (model 1).

**Table 14 materials-12-02228-t014:** Statistical evaluation of *H:C* ratio model of torrefied pruned biomass (model 2).

Intercept/Coefficient	Value of Intercept/Coefficient	Standard Error	*p*	Lower Limit of Confidence	Upper Limit of Confidence	Standardized *β* Coefficient
*a* _1_	2.207455	0.044156	0.000000	2.120162	2.294748	−
*a* _2_	0.000394	0.000029	0.000000	0.000338	0.000451	1.68
*a* _3_	−0.000153	0.000000	0.000000	−0.000153	−0.000153	−2.15

*H:C* = *a*_1_ + *a*_2_·*t*^2^ + *a*_3_·*T*·*t*, R^2^ = 0.82, (model 2), *AIC* = 200.

**Table 15 materials-12-02228-t015:** Statistical evaluation of *O:C* ratio model of torrefied pruned biomass.

Intercept/Coefficient	Value of Intercept/Coefficient	Standard Error	*p*	Lower Limit of Confidence	Upper Limit of Confidence	Standardized *β* Coefficient
*a* _1_	0.022935	0.005095	0.000000	0.012860	0.033010	-
*a* _2_	−0.000114	0.000040	0.000000	−0.000192	−0.000035	−3.03
*a* _3_	1.85 × 10^−5^	0.000000	0.000000	0.000000	0.000000	2.47
*a* _4_	−0.000047	0.000049	0.000000	−0.000143	0.000050	−0.59
*a* _5_	5.38 × 10^−5^	0.000000	0.000000	0.000001	0.000001	0.55
*a* _6_	−6.26 × 10^−5^	0.000000	0.000000	0.000000	0.000000	−0.21

*O:C* = *a*_1_ + *a*_2_·*T* + *a*_3_·*T*^2^ + *a*_4_·*t* + *a*_5_·*t*^2^ + *a*_6_·*T*·*t*, R^2^ = 0.48.

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
