# Peer review of "Oxytree Pruned Biomass Torrefaction: Mathematical Models of the Influence of Temperature and Residence Time on Fuel Properties Improvement"

_materials, 2019, doi:10.3390/ma12142228_

Round 1
Reviewer 1 Report
This work shows regression modeling and model interpretation of several parameters from torrefaction of Paulownia prunings, along with cost analysis of the resulting biomass. The manuscript contains useful information and the quality of English is good but several points shown below need to be addressed to make this manuscript better.
“To date, ~10 % of energy)” total primary energy supply TPES?
It would be good to write a sentence or two to highlight your recent data article in the Introduction section.
Fig. 2 - update caption to describe S(G)(I) etc. Where are N+ and N- notations in the figure?
Define standardized coefficient in methods section. Is this same as Pearson’s?
Fig. 3 – First, check if the journal is okay with this style – you have a full-fledged table inside of a figure. Usually they are separate. Write the full p-value 10^-x etc. just 0.00 doesn’’t show the extent of significance. Limit of confidence- how much confidence? Is it 95%?
Fig. 3 – how is it that your value of coefficients a3 and a6 exactly matches the upper or lower confidence limit? Shouldn’t the coefficient value be somewhere between the LCL and UCL but not exactly at the LCL or UCL value?
What are the blue circles in figures?
Fig. 3 – “The greatest positive impact on MY was associated with T (β = 0.83) and t2 (β = 0.56). This positive impact was reduced by predictor T2 (β = -1.49), t (β = -0.19), and T∙t (β = -0.74), respectively.” Fig. 4 – “The EY was affected by predictor T (β = -2.35) and T∙t (β = -1.16). Because the predictor T2 has the highest negative value, it is reasonable to assume that torrefaction temperature has the biggest impact on decreasing the EY.” Your reasoning for impact of independent variable for MY vs EY is confusing and contradictory. For MY, T-square has the largest value but you wrote that impact of first order T was reduced by second order T. For EY, you wrote that T2 has the biggest impact but EY was affected by T. -2.35 is the beta value for T-squared regression coefficient a3 instead, as shown in Fig. 4. All I see from MY and EY regressions is that T-squared is a better predictor of MY and EY. Now the positive values of first order T and negative values of second order T are confusing. If you forget about the outcomes of statistical analysis and just look the MY and EY figures, they clearly show that mass and energy yields decrease with increasing temperature, which is what one would expect. Therefore, the regression coefficient that shows negative sign for temperature, which in this model is T-square, seems to describe the temperature relationship better. But your sentence that says “The greatest positive impact on MY was associated with T (β = 0.83)” gives the impression that MY and EY increase with temperature, which is clearly not what the graph shows. Check the manuscript again for such errors and revise the relevant text for better explanations. Check such instances for the time variable as well.
“concluded that temperature had a greater impact on EDr improvement than the process time.” The outcome of EDr graph is incomplete. Overall, the largest coefficient a3 and a4 suggest that higher temperature and higher time result in biochar with higher energy density. Please check for other figures.
Explain in details what is Akaike analysis, how it is applied and what is the significance of low or high value. In the equation, does ln mean natural log? Why have you applied AIC to only a few of the dependent variables like LHV but not MY, EY, EDr etc.?
Fig. 14 for sulfur content looks very odd. One would expect that with decreasing mass yield, S% would increase. Is it because of lack of precision in measurement or oxidation and release as a gas during torrefaction?
“the best conditions to torrefaction process were T = 200 °C, and t = 20 min.” This conclusion and the graphs clearly show that there is room for improving torrefaction performance of oxytree prunings. These were the mildest T and t in this work and suggests that lower temperature or time will further improve it. But it is kind of surprising as these conditions seem to be quite mild. It seems to me that the pruinings have lot of leaf matter which maybe easier to breakdown than woody stem material. What was the % leaves vs. stem dry matter in the prunings.
It would be better if you highlight more that these are Paulownia prunings, specially in Introduction and Conclusions, so that readers are clear about this fact. Did you write about prunings and methodology in your earlier data article?
“The highest HHV of torrefied biomass was 21 MJ·kg-1, at 300 °C and 20 min. The study found that the most beneficial economic aspect parameters of torrefaction are 200 °C and 20 min” Write the HHV at 200C for 20 min. The HHV graph shows that at 200C and 20 min, HHV is around 18 MJ/kg. Are you saying that it is more economical beneficial to do torrefaction at 200C and 20 min despite lower HHV in these milder conditions?
Author Response
We submit our responses to the reviewer comments in the attached file.

Reviewer 2 Report
The subject is interesting and actual. It is a subject with direct application and can be very useful research for those who need to treat agriculture or forestry waste using torrefaction as pre-processing technology.
The overall classification of this article is good.
Concerning recommendations to help the improvement of the final quality of the article, I can recommend improving significatively the quality of the figures. Try to explain better the content of the figures in plain text.
The same recommendation for the tables, in particular to those in the final part of the article.
Authors should try to check for actual references, and some of the most important are missing. Some authors must mandatorily be cited in an article like this.
In Figure 18, the authors designated torrefaction final product as biochar. This is not correct. Biochar is the product of carbonization or slow pyrolysis. In this case, author must designate torrefied material as it is, torrefied biomass.
Author Response

(The authors gave the same response as above.)

Round 2
Reviewer 1 Report
Authors made all the necessary changes and answered all of the comments. This manuscript can now be accepted for publication.